# Effects on NPK Status, Growth, Dry Matter and Yield of Rice (*Oryza sativa*) by Organic Fertilizers Applied in Field Condition

**Kyi Moe** [1,2,*] **, Aung Zaw Htwe** [1,2] **, Thieu Thi Phong Thu** [1] **, Yoshinori Kajihara** [3] **and Takeo Yamakawa** [4]

[1] Plant Nutrition Laboratory, Graduate School of Bioresource and Bioenvironmental Sciences, Faculty of Agriculture, Kyushu University, 744 Motooka Nishi-ku, Fukuoka 819-0395, Japan; aungzawhtwe333@gmail.com (A.Z.H.); phongthu_nha84@yahoo.com (T.T.P.T.)

[2] Department of Agronomy, Yezin Agricultural University, Yezin, Nay Pyi Taw 15013, Myanmar

[3] University Farm, School of Agriculture, Kyushu University, Fukuoka 812-8581, Japan; kajihara@farm.kyushu-u.ac.jp

[4] Plant Nutrition Laboratory, Division of Molecular Biosciences, Department of Biosciences & Biotechnology, Faculty of Agriculture, Kyushu University, 744 Motooka Nishi-ku, Fukuoka 819-0395, Japan; yamakawa@agr.kyushu-u.ac.jp

* Correspondence: kyimoeyau@gmail.com; Tel.: +81-092-642-2847

**Abstract:** The decline in rice yields as a result of excessive chemical fertilizer (CF) inputs is a matter of great concern in rice-growing regions of Asia. In two-year's field experiments, the nitrogen, phosphorus, and potassium (NPK) status, growth characteristics and yield of rice were examined by application of poultry manure (PM), cow manure (CM) and compost (CP). Organic fertilizers were applied as EMN (estimated mineralizable N) based on their total N content. Six treatments were assigned in a randomized complete block design: (1) no-N fertilizer ($N_0$); (2) 50% CF ($CF_{50}$), (3) 100% CF ($CF_{100}$); 50% CF + 50% EMN from (4) PM or (5) CM or (6) CP. Compared with $CF_{100}$, the $CF_{50}PM_{50}$ (total N ≥ 4%) accumulated higher N, P and K content in leaf, sheath, panicle and seeds, resulting in greater growth and yield. The $CF_{50}PM_{50}$ increased yield by 8.69% and 9.70%, dry matter by 4.76% and 5.27% over $CF_{100}$ in both years. The continuous application of $CF_{50}CM_{50}$ (total N < 4%) and $CF_{50}CP_{50}$ (total N < 4%) treatments led to similar NPK contents but higher yields than those of $CF_{100}$ treatment in 2018. In conclusion, the organic fertilizer (total N ≥ 4%) with the EMN method enhances higher N availability in each year. Continuous application of organic fertilizer (total N < 4%) over two years effectively increased N availability in the second year. The 50% organic fertilizer (total N ≥ 4%) and 50% CF led to increased NPK availability and rice yields over the 100% CF treatment, reducing CF usage and leading for sustainable agriculture.

**Keywords:** chemical fertilizer; NPK content; organic fertilizer; rice; yield

## 1. Introduction

More than 759.6 Mt of rice (*Oryza sativa* L.) was produced globally in 2017 [1]. Approximately 90% of annual production is grown and consumed in Asia. However, mean yields in Asia are low compared to global mean yields [2]. Nevertheless, there are several ways to increase crop yields. For example, the proper management of nitrogen (N) fertilizers is important for improving rice yields [3]. In general, fertilizers containing N, phosphorus (P), and potassium (K), essential plant nutrients, are vital for productive crops [3,4].

Although crop production requires fertilizers, the overly large doses and use of fertilizers with chemically unbalanced NPK ratios and in intensive rice production has resulted in soil-related problems, such as acidification [5], loss of organic matter, deterioration of the structure, and reductions in biological activities and fertility [6]. Consequently, crop yields in several regions are stagnating or declining [7]. Modern, high-yield rice varieties have a high nutrient demand and are very responsive to fertilizer inputs. However, these varieties also mine the soil of nutrients at higher rates than traditional varieties [8]. It is difficult to maximize the yields of crops grown in degraded soil, and the effort required to mitigate degraded soil is unsustainable on the global scale.

In recent times, farmers have mostly relied on chemical fertilizers (CFs), particularly N fertilizers, to boost rice yields. Farmers in Asia have also applied less amount of P and K fertilizers [9]. Initially, rice yields were increased by applying large amounts of CFs. However, this has led to soil problems, declining crop yields, and global environmental issues. Thus, we need to develop and adopt environmentally friendly alternatives that can supplement or replace CFs. Organic fertilizers are environmentally sustainable and can maintain soil health when used in intensive rice agriculture. They help to conserve the amount and quality of organic matter in the soil, and supply N, P, K, and essential micronutrients [10,11]. For example, the use of organic manure can lead to significant increases in levels of organic matter, EDTA-extractable iron, zinc, copper, and plant-available N, P, and K in the soil [12].

Generally, farmers apply manure by weight without considering total N content or the percentage of mineralizable N in the manure. The N transformation is different between organic manure and inorganic fertilizer. When manure is applied to the soil, a portion of the organic N is converted into ammonium-N ($NH_4$-N) by soil microbes and the $NH_4$-N is then converted into nitrate by nitrifying soil microbes. Plants can only use mineralized nutrients. Therefore, organic fertilizer application should be based on mineralizable N content rather than total fertilizer weight to fulfill crop N demand. The percentage of mineralizable N in organic fertilizer depends on total N content; in Nishio [13], manure with a total N content <2%, 2%–4%, and ≥4% released 20%, 30%, and 50% of the total N as mineralizable N, respectively, in the first year after application. In this study, we applied poultry manure (PM), cow manure (CM), and compost (CP), which was made from kitchen waste and bamboo powder, based on levels of estimated mineralizable N (EMN; Table 1). The EMN for each fertilizer type was calculated according to the relationship between the amount of mineralizable N and their total N in Nishio [13]. We hypothesized that the application of organic fertilizers according to EMN content would be a promising technique to fulfill crop N needs.

Despite their advantages, using only organic fertilizers is not efficient as they have a low nutrient content compared to CFs. Plants grown with organic fertilizer alone may suffer from nutrient deficiencies and produce low yields. However, the application of organic manure together with CFs helps neutralize soil pH, and leads to higher levels of organic carbon and improved macro- and micronutrient availability, physical properties, and microbial activity [14], thereby increasing crop yields [15–17].

We investigated how to synchronize mineralizable N availability to crop N demand and supply sufficient P and K by applying organic fertilizers based on EMN contents. We analyzed the N, P, and K contents of leaves, sheaths, panicles, and seeds of rice plants throughout the crop period. Previous studies have mostly focused on the nutrient contents of rice grain and straw at harvest. Conversely, few studies on rice have reported on nutrient contents in specific plant parts at all critical growth stages, particularly while using experiments based on the EMN contents of organic fertilizers.

The application of organic fertilizers based on EMN contents together with CFs should synchronize N supply and crop demand. In addition, this delivery method should supply sufficient P and K to meet crop needs, which would facilitate better plant development at all growth stages and enhance crop yields. We also investigated whether the co-application of organic fertilizers and CFs would improve crop growth and yield compared to only using CFs.

## 2. Materials and Methods

### 2.1. Description of Experimental Site

Field experiments were conducted at the Kyushu University farm in Fukuoka Prefecture, Japan (33°37'N, 130°25'E), from June to October in both 2017 and 2018. The mean maximum and minimum temperatures from June to October for both years were 35.0 °C and 17.1 °C, respectively, whereas minimum and maximum monthly precipitation were 23.5 mm and 289.5 mm, respectively. The soil texture at this site was classified as clay loam and had a $pH_{H2O}$ of 6.12. The soil contained 0.15% total N, 5.20 mg N 100 $g^{-1}$ mineralizable N, 0.25% phosphorus pentoxide ($P_2O_5$), 15.37 mg P 100 $g^{-1}$ available P, and 0.48% potassium oxide ($K_2O$). In addition, the medium of exchangeable K for the soil was 0.37 $cmol_c$ $kg^{-1}$ and its cation exchangeable capacity (CEC) was 15.70 $cmol_c$ $kg^{-1}$).

### 2.2. Experimental Design and Treatment Application

We used a randomized complete block design with three replicates for each treatment. Six treatments were assigned: (1) no nitrogen fertilizer ($N_0$; control); (2) 50% CF ($CF_{50}$); (3) 100% CF ($CF_{100}$); and 50% CF plus (4) PM with 50% EMN ($CF_{50}PM_{50}$); (5) CM with 50% EMN ($CF_{50}CM_{50}$); or (6) CP with 50% EMN ($CF_{50}CP_{50}$). We used Keifun PM and Gyufun CM, and the CP was made from kitchen waste and bamboo powder. For the $CF_{100}$ treatment, 85 kg N $ha^{-1}$ was applied in the form of urea, whereas 42.5 kg $ha^{-1}$ urea was applied for the $CF_{50}$ treatment, corresponding to 50% EMN. EMN levels for PM, CM, and CP (Table 1) were calculated according to the relationship between the amount of mineralizable N and their total N in Nishio [13]. In manure, the amount of mineralizable N mainly depends on total N content. Thus, manure containing < 2%, 2%–4%, or ≥ 4% total N release will have 20%, 30%, or 50% mineralizable N, respectively.

$$\text{EMN}\left(\text{kg ha}^{-1}\right) = \text{Wt. organic fertilizer (DW)}\left(\text{kg ha}^{-1}\right) \times \frac{\text{Total N (\%)}}{100} \times \frac{\text{Mineralizable N (\%)}}{100}.$$

Our experimental area was initially irrigated to facilitate disc plowing and harrowing. Subsequently, plots were lined with plastic to a depth of 15 cm to demarcate plot boundaries and prevent seepage between adjacent plots. Large plastic liners were installed to separate treatment replicates. We applied the same treatment to each plot in both years to observe the continuous effects of the treatments.

**Table 1.** Weight and nitrogen (N), phosphorus (P) and potassium (K) applied (kg $ha^{-1}$) from organic and inorganic fertilizers.

| No. | Treatments | Organic Fertilizers | | | | | Inorganic Fertilizers | | |
|---|---|---|---|---|---|---|---|---|---|
| | | DW | N | $P_2O_5$ | $K_2O$ | EMN | N | $P_2O_5$ | $K_2O$ |
| 1 | $N_0$ | - | - | - | - | - | 0.0 | 60.0 | 85.0 |
| 2 | $CF_{50}$ | - | - | - | - | - | 42.5 | 60.0 | 85.0 |
| 3 | $CF_{100}$ | - | - | - | - | - | 85.0 | 60.0 | 85.0 |
| 4 | $CF_{50} PM_{50}$ | 1745.3 | 85.0 | 79.5 | 37.3 | 42.5 | 42.5 | 60.0 | 85.0 |
| 5 | $CF_{50} CM_{50}$ | 5927.4 | 141.6 | 113.2 | 90.1 | 42.5 | 42.5 | 60.0 | 85.0 |
| 6 | $CF_{50} CP_{50}$ | 6558.6 | 141.6 | 148.8 | 93.1 | 42.5 | 42.5 | 60.0 | 85.0 |

Subscript numbers of treatments show the amount of N applied as a percentage based on 85 kg N (or EMN) $ha^{-1}$. EMN = estimated mineralizable N, PM = poultry manure, CM = cow manure, CP = compost, DW = dry weight.

### 2.3. Application of Organic Fertilizers and CFs

PM, CM, and CP were incorporated into the soil 2 days before the rice was transplanted. No N fertilizers were applied to control plots ($N_0$), but they did receive P and K fertilizers. All plots received 60 kg $ha^{-1}$ $P_2O_5$ (as superphosphate) and 85 kg $ha^{-1}$ $K_2O$ (as muriate of potash) as CFs. Urea and muriate of potash were applied at three doses: 60% was incorporated into the soil 1 day before

transplantation, 20% in the active-tillering stage, and the remaining 20% in the panicle-initiation stage. All of the superphosphate was applied 1 day before transplantation.

## 2.4. Sampling and Soil Analyses

Before field experiments commenced, soil samples to depths of 15 cm were taken using a tube with a diameter of 5 cm from nine locations in the field. For each sample, Soil $pH_{H2O}$ (1:2.5 soil: $H_2O$) was measured using a pH meter (HM-10P; DKK-TOA Corp., Tokyo, Japan). Soil samples were also digested using the salicylic acid–sulfuric acid–hydrogen peroxide method [18]; then, total N was analyzed using the indophenol method [19], and total P was tested using the ascorbic acid method [20]. Cation exchange capacity and exchangeable cations were determined using ammonium acetate extraction with shaking [21]. To analyze mineralizable N, we first used the soil incubation method [22] followed by the indophenol method [19]. Available P was analyzed using Truog's method [23] followed by the ascorbic acid method [20].

## 2.5. Analyses of Organic Fertilizers

Total N and P contents in the organic fertilizers were analyzed with the same methods used for soil analyses. Total K, calcium (Ca), and magnesium (Mg) contents (Table 2) were analyzed using an atomic absorption spectrophotometer (Z-5300; Hitachi, Tokyo, Japan) after samples were digested. Finally, available N, comprising $NH_4$-N and nitrate-N ($NO_3$-N), and sodium (Na) were extracted from the organic fertilizers using the hot water extraction method [24].

**Table 2.** Chemical compositions of poultry manure, cow manure and compost.

| No. | Sample | C:N | Moisture (%) | Total % (Dry Basis) | | | | | | | |
|-----|--------|-----|--------------|-----|--------|--------|----------|--------|------|------|------|
| | | | | N | $NH_4$-N | $NO_3$-N | $P_2O_5$ | $K_2O$ | Ca | Mg | Na |
| 1. | Poultry manure | 6.80 | 21.86 | 4.87 | 0.75 | 0.00 | 4.56 | 2.14 | 8.85 | 0.75 | 0.28 |
| 2. | Cow manure | 15.23 | 46.40 | 2.39 | 0.19 | 0.00 | 1.91 | 1.52 | 1.09 | 0.55 | 0.34 |
| 3. | Compost | 17.71 | 41.02 | 2.16 | 0.04 | 0.02 | 2.27 | 1.18 | 5.28 | 0.21 | 0.85 |

## 2.6. Crop Management

The Manawthukha rice (*Oryza sativa* L.) variety (Indica type, high-yielding Myanmar variety) was cultivated for this study. Seeds were obtained from a research laboratory at Kyushu University [9]. This rice variety is well-adapted to field conditions in Japan. Viable seeds were selected using the sodium chloride (NaCl) solution method, with the specific gravity of the solution set to 1.08 [25]. Then they were washed three times with deionized water, sterilized by shaking in 10% ethanol at 150 rpm for 3 min, and washed again at least three more times in deionized water. Thereafter, the seeds were shaken in 5% sodium hypochlorite solution at 150 rpm for 30 min. Sterilized seeds were washed in deionized water and germinated in the dark in an incubator at 25 °C for 48 h.

Germinated seeds were sown homogenously on seedbeds (100 g/tray) prepared using commercial seedbed soil (Kokuryu Baido; Seisin Sangyo Co., Kitakyushu, Japan). Once seedlings were 21 days old (June 22, 2017, and June 21, 2018), they were transplanted to hills spaced 25 × 15 cm apart. Two seedlings were planted per hill. The experimental field was irrigated using the same system as surrounding university farms. Rice plants were harvested 125 days after transplanting (DAT), on October 26 for 2017 and October 25 for 2018.

## 2.7. Measurement of Plant Growth Characteristics

We marked five representative hills within each experimental plot with poles to measure plant height (cm), and also recorded the number of tillers per hill and used the soil–plant analysis development (SPAD) tool to measure chlorophyll content and/or N concentration in the leaves throughout the crop period. These growth characteristics were measured weekly from 10 DAT to the 50% flowering, and

then at 2-week intervals. The SPAD value of the uppermost fully expanded leaf was measured using the SPAD-502 chlorophyll meter (Konica Minolta, Inc., Osaka, Japan) before panicle initiation, and the SPAD value of the flag leaf was measured afterward.

## *2.8. Determination of Dry Matter Production, Nutrient Content, Yield, and Yield Characteristics*

At the stages of active tillering, panicle initiation, flowering, and harvest, all plants were cut just above the ground. The plants were divided into sheaths, leaves, panicles, and seeds. Each plant part was oven-dried separately at 70 °C for 48 h and then immediately weighed. Dry matter (DM) accumulation was determined by summing the dry weights of sheaths, leaves, panicles, and seeds, and expressed as metric tons per hectare (t ha$^{-1}$).

Sheath, leaf, panicle, and seed samples were digested separately using the salicylic acid–sulfuric acid–hydrogen peroxide method [18]. N, P, and K contents were analyzed as previously described. At harvest, the total accumulation of each nutrient was calculated by summing the product of the biomass and concentration of that nutrient in each plant part.

Ten hills within each plot were also sampled at harvest to measure yield and yield characteristics such as number of panicles per hill, number of spikelets per panicle, percentage of filled grains, thousand grain weight (g), and maximum panicle length (cm). Filled and unfilled seeds were separated using a wind blower machine. Then the filled seeds were immersed in an NaCl solution with a specific gravity of 1.08 to separate heavy seeds from light seeds, after which the seeds were washed three times with deionized water. Rice yield was determined using the weight of the heavy seeds adjusted to 14% moisture content. The harvest index (HI) was calculated as the ratio of economic yield (total seed weight) to biological yield (total DM weight) [25].

## *2.9. Statistical Analyses*

Data were compared among treatments using one-way analysis of variance. Then, the treatment means were subjected to pairwise comparisons with the Tukey's honestly significant difference test using Statistix software (version 8.0; Analytical Software, Tallahassee, FL, USA). Differences were considered statistically significant at $p < 0.05$.

## 3. Results

### *3.1. Plant Growth Characteristics*

Plant heights followed a similar trend in each year. Heights were not different among treatments before 30 DAT (Figure 1). After the active tillering stage (38 DAT), plants in the $CF_{50}PM_{50}$ treatment were taller than those in other treatments ($p < 0.05$). At harvest time, the maximum plant heights of 109.76 cm and 107.2 cm were observed in the $CF_{50}PM_{50}$ treatment in 2017 and 2018, respectively ($p < 0.01$). Plants in the $CF_{50}CM_{50}$ and $CF_{50}CP_{50}$ treatments were short during the growth stages and similar in size to those in the $CF_{100}$ treatment in both years.

In both years, tiller numbers increased rapidly in all treatments from 20 to 30 DAT (Figure 2) but were not significantly different among treatments. Nevertheless, higher tiller numbers were observed in the $CF_{50}PM_{50}$ treatment across all growth stages. The maximum tiller number was observed at 40 DAT; 34.73 and 38.22 tillers per hill in the $CF_{50}PM_{50}$ plots in 2017 and 2018, respectively ($p < 0.01$). Tiller numbers declined after 45 DAT in all treatments. At harvest time, the maximum tiller numbers of 18.2 and 22.5 tillers per hill were observed in $CF_{50}PM_{50}$ plots in 2017 and 2018, respectively. Tiller numbers in the $CF_{100}$, $CF_{50}CM_{50}$, and $CF_{50}CP_{50}$ treatments were not significantly different in both years.

SPAD values increased rapidly in all treatments from 0 to 17 DAT (Figure 3). Then they decreased slightly until 40 DAT (maximum tillering stage) and increased again from 59 DAT (panicle-initiation stage) in all treatments. During the grain-filling period, the maximum SPAD values of 36.15 and 36.59 were observed in the $CF_{50}PM_{50}$ treatment in 2017 and 2018, respectively ($p < 0.01$). The values

decreased gradually until maturity. Primarily, the SPAD values were consistently higher in the $CF_{50}PM_{50}$ treatment across all growth stages in both years and were similarly low in the $CF_{50}CM_{50}$, $CF_{50}CP_{50}$, and $CF_{100}$ treatments.

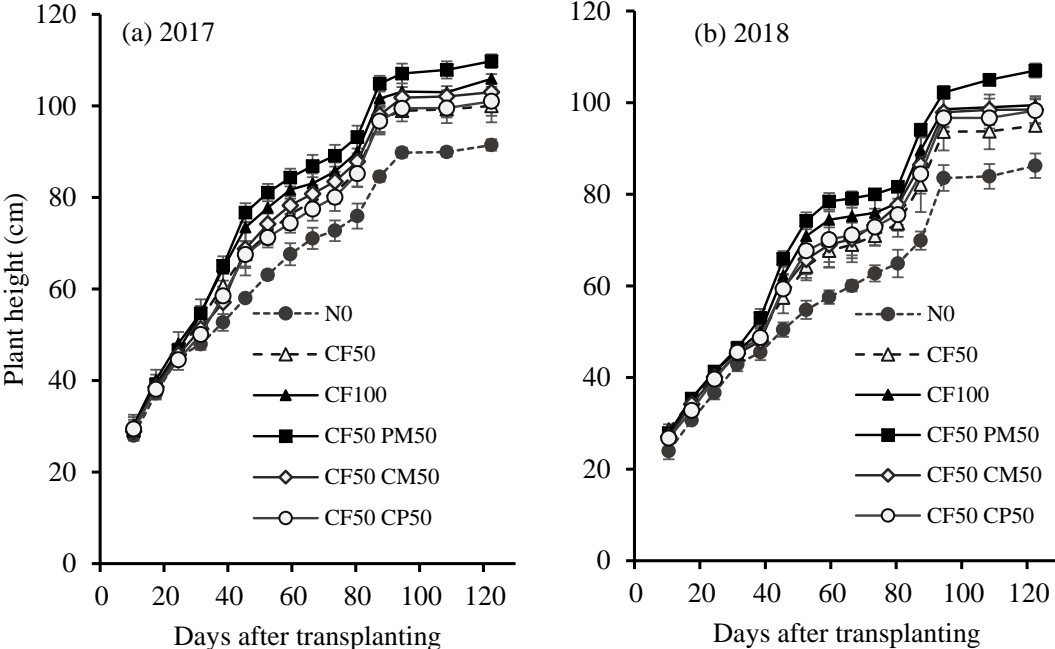

**Figure 1.** Plant height (cm) of rice variety Manawthukha affected by organic fertilizers. The numbers followed by treatment show the amount of N applied as a percentage based on 85 kg N (or estimated mineralizable nitrogen; EMN) ha$^{-1}$. CF = chemical fertilizer, PM = poultry manure, CM = cow manure, CP = compost. Vertical bars represent the SD of three samples.

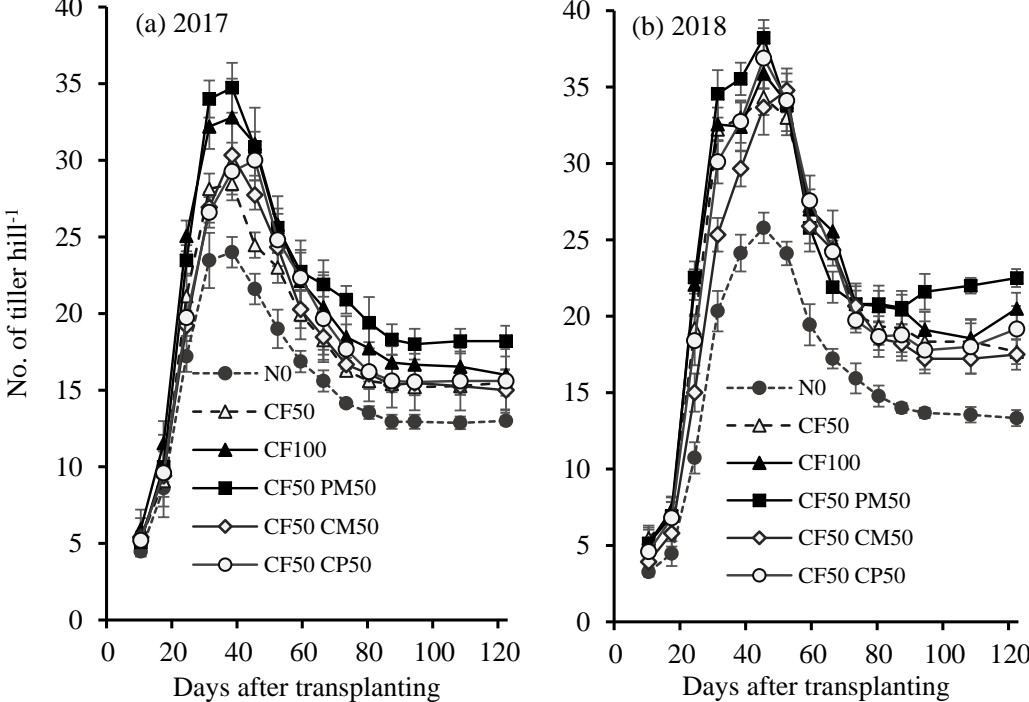

**Figure 2.** Tiller number of rice variety Manawthukha affected by organic fertilizers. Vertical bars represent the SD of three samples.

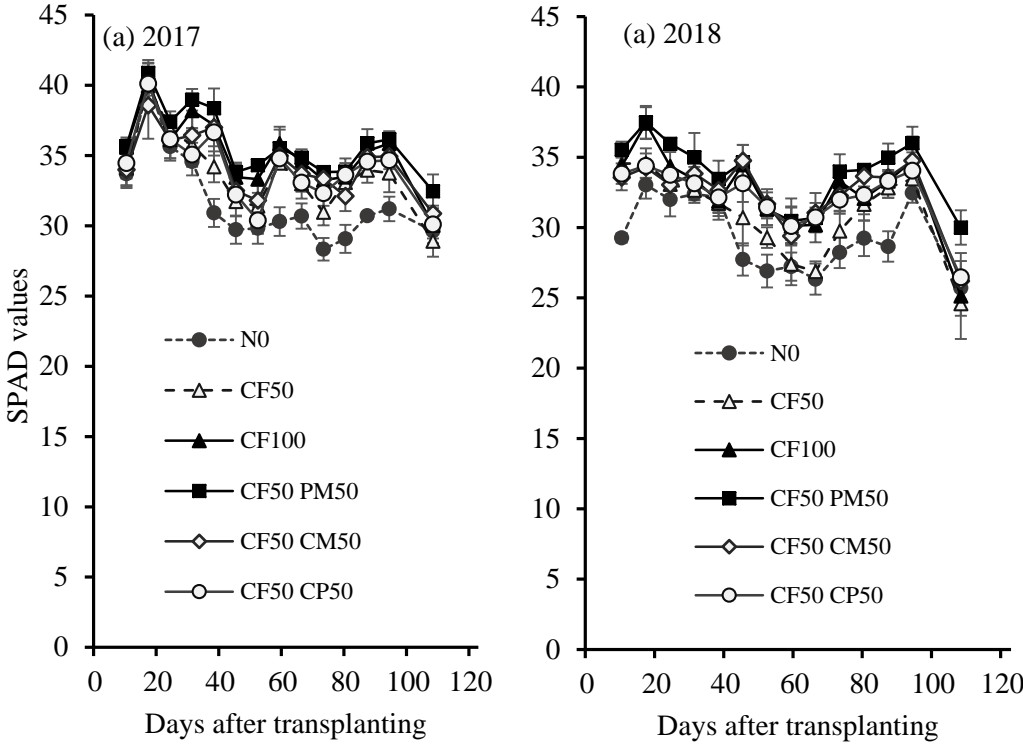

**Figure 3.** Changes in soil–plant analysis development (SPAD) values of rice variety Manawthukha affected by organic fertilizers. Vertical bars represent the SD of three samples.

Rice plants accumulated a large amount of DM during the tillering stage in the $CF_{100}$ treatment in 2017 (Figure 4). After that stage, DM production was similar among the $CF_{100}$, $CF_{50}PM_{50}$, $CF_{50}CM_{50}$, and $CF_{50}CP_{50}$ treatments. At harvest, the maximum DM produced was 15.01 t ha$^{-1}$, from the $CF_{50}PM_{50}$ treatment, followed by 14.29 t ha$^{-1}$ in the $CF_{100}$ treatment. The amounts of DM produced in the $CF_{50}CM_{50}$ and $CF_{50}CP_{50}$ treatments were 12.30 and 11.26 t ha$^{-1}$, respectively. The most DM was produced in $CF_{50}PM_{50}$ plots in 2018 even during the early stage ($p < 0.05$). Throughout the crop period, plots treated with organic fertilizers produced DM amounts similar to those from plots treated with $CF_{100}$. At harvest time, $CF_{50}PM_{50}$ plots produced the most DM, 15.49 t ha$^{-1}$ ($p < 0.01$). $CF_{50}CM_{50}$ and $CF_{50}CP_{50}$ also produced significantly more DM in 2017 than in 2018. DM production in the $CF_{50}CM_{50}$ and $CF_{50}CP_{50}$ treatments was not significantly different from that in the $CF_{100}$ treatment.

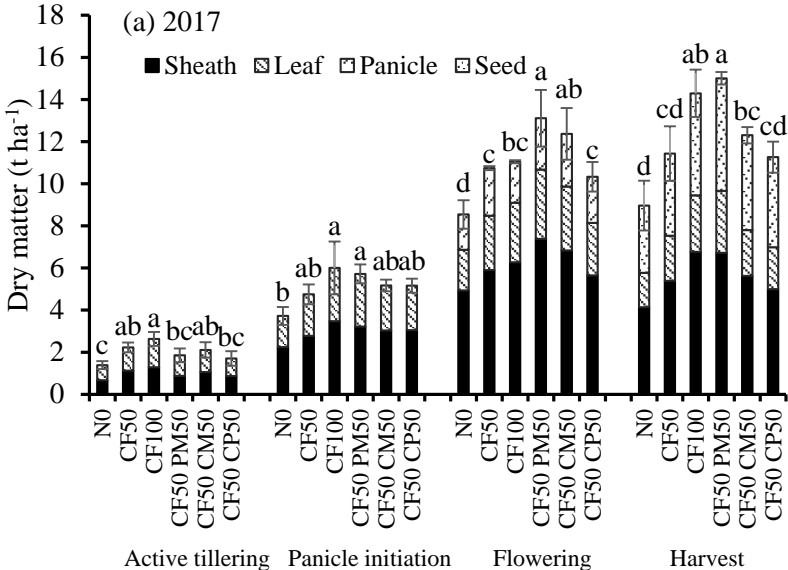

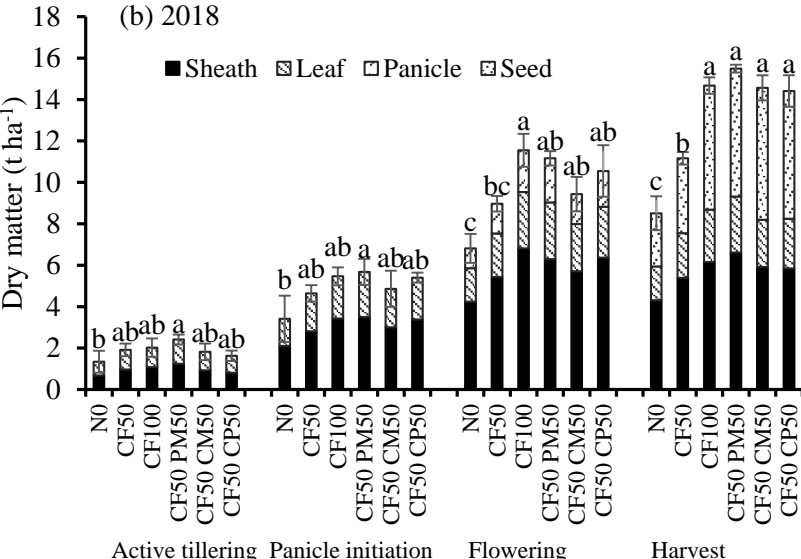

**Figure 4.** Dry matter (t ha$^{-1}$) of rice variety Manawthukha affected by organic fertilizers at the critical growth stages. The histograms with the same letter in the same case at each stage are not significantly different by the Tukey honestly significant difference (HSD) test ($p < 0.05$). Vertical bars represent the SD of three samples.

### 3.2. N, P, and K Contents in Plant Parts at Critical Growth Stages

Plants treated with CF$_{50}$PM$_{50}$ had the highest leaf and sheath N contents at the tillering, panicle-initiation, flowering, and harvest stages in both years ($p < 0.05$; Table 3). Panicle N content was similar for plants treated with CF$_{50}$PM$_{50}$ and CF$_{100}$ in 2017, whereas it was higher in CF$_{50}$PM$_{50}$ plants than in CF$_{100}$ plants in 2018 ($p < 0.05$). CF$_{50}$PM$_{50}$ plants also had the highest seed N contents in 2017 (1.27%) and 2018 (1.48%). CF$_{100}$ plants had high levels of N in leaves, sheaths, and seeds at the tillering, panicle-initiation, and harvest stages in both years, but these levels were still less than those of CF$_{50}$PM$_{50}$ plants. CF$_{50}$CM$_{50}$ and CF$_{50}$CP$_{50}$ plants had low levels of N in all plant parts in all growth stages in 2017, but these levels increased in 2018 to levels similar to those in the CF$_{100}$ treatment.

CF$_{50}$PM$_{50}$ plants had the highest leaf and sheath P contents in all growth stages in both years ($p < 0.05$; Table 3). The highest panicle P content was observed in CF$_{50}$PM$_{50}$ plants, but this value was

not significantly different from those in $CF_{100}$, $CF_{50}CM_{50}$, and $CF_{50}CP_{50}$ plants. Seed P contents of $CF_{50}PM_{50}$ plants were significantly higher than those for other treatments, at 0.27% and 0.28% in 2017 and 2018, respectively. Plants grown with CM and CP had higher seed P contents in 2018 than in 2017.

In both years, $CF_{50}PM_{50}$ plants had the highest leaf and sheath K content in the tillering, panicle initiation, flowering, and harvest stages ($p < 0.05$; Table 3). However, panicle K content did not differ among treatments. $CF_{50}PM_{50}$ plants had the highest seed K contents overall, at 0.36% and 0.32% in 2017 and 2018, respectively. In 2018, leaf, sheath, panicle, and seed K contents of plants treated with organic fertilizers were significantly higher than those of plants treated with $CF_{100}$.

**Table 3.** N, P and K contents (%) of leaf, sheath, panicle and seeds of rice variety Manawthukha affected by organic fertilizer at the critical growth stages.

| Treatment | Active Tillering | | Panicle Initiation | | Flowering | | | Harvest | | |
|---|---|---|---|---|---|---|---|---|---|---|
| | Leaf | Sheath | Leaf | Sheath | Leaf | Sheath | Panicle | Leaf | Sheath | Seed |
| 2017 N (%) | | | | | | | | | | |
| $N_0$ | 2.84 b | 1.22 b | 1.55 c | 0.62 c | 0.84 b | 0.45 c | 1.08 c | 0.38 c | 0.52 b | 1.18 d |
| $CF_{50}$ | 2.93 b | 1.07 b | 1.67 bc | 0.75 bc | 0.84 b | 0.50 bc | 1.11 c | 0.45 bc | 0.62 ab | 1.24 bc |
| $CF_{100}$ | 3.27 b | 1.26 ab | 1.86 ab | 0.86 ab | 0.95 ab | 0.57 a | 1.24 a | 0.50 ab | 0.71 a | 1.31 a |
| $CF_{50}PM_{50}$ | 3.87 a | 1.50 a | 1.95 a | 0.93 a | 1.05 a | 0.57 a | 1.20 ab | 0.56 a | 0.72 a | 1.27 ab |
| $CF_{50}CM_{50}$ | 3.09 b | 1.07 b | 1.73 bc | 0.82 ab | 0.92 ab | 0.52 ab | 1.15 bc | 0.47 b | 0.63 ab | 1.22 bcd |
| $CF_{50}CP_{50}$ | 2.87 b | 1.00 b | 1.61 c | 0.74 bc | 0.94 ab | 0.51 abc | 1.14 bc | 0.44 bc | 0.61 ab | 1.20 cd |
| 2018 N (%) | | | | | | | | | | |
| $N_0$ | 2.93 a | 1.17 a | 1.70 c | 0.65 b | 1.02 b | 0.46 a | 1.18 a | 0.47 bc | 0.42 b | 1.22 c |
| $CF_{50}$ | 2.79 b | 1.10 ab | 1.69 d | 0.66 b | 1.02 b | 0.45 ab | 1.10 a | 0.43 c | 0.45 b | 1.24 c |
| $CF_{100}$ | 2.84 ab | 1.08 b | 1.79 ab | 0.67 ab | 1.04 ab | 0.44 b | 1.07 a | 0.53 ab | 0.59 a | 1.40 ab |
| $CF_{50}PM_{50}$ | 2.93 a | 1.12 ab | 1.83 a | 0.74 a | 1.05 ab | 0.46 a | 1.08 a | 0.59 a | 0.59 a | 1.46 a |
| $CF_{50}CM_{50}$ | 2.75 b | 1.07 b | 1.75 b | 0.66 b | 1.16 a | 0.47 a | 1.15 a | 0.50 bc | 0.48 ab | 1.28 bc |
| $CF_{50}CP_{50}$ | 2.82 ab | 1.13 ab | 1.80 a | 0.68 ab | 1.09 a | 0.45 ab | 1.10 a | 0.53 ab | 0.52 ab | 1.41 ab |
| 2017 P (%) | | | | | | | | | | |
| $N_0$ | 0.25 a | 0.25 bc | 0.20 a | 0.23 b | 0.08 b | 0.13 a | 0.20 b | 0.03 c | 0.03 c | 0.26 ab |
| $CF_{50}$ | 0.24 ab | 0.24 c | 0.19 ab | 0.24 ab | 0.08 b | 0.13 a | 0.20 ab | 0.04 abc | 0.04 abc | 0.26 ab |
| $CF_{100}$ | 0.25 ab | 0.26 ab | 0.19 ab | 0.25 ab | 0.08 ab | 0.13 a | 0.22 a | 0.05 ab | 0.05 ab | 0.26 ab |
| $CF_{50}PM_{50}$ | 0.27 a | 0.27 a | 0.19 ab | 0.26 a | 0.09 a | 0.15 a | 0.22 ab | 0.05 a | 0.05 a | 0.27 a |
| $CF_{50}CM_{50}$ | 0.24 b | 0.25 c | 0.18 b | 0.25 ab | 0.08 b | 0.13 a | 0.21 ab | 0.04 abc | 0.04 abc | 0.25 c |
| $CF_{50}CP_{50}$ | 0.24 ab | 0.24 c | 0.19 ab | 0.24 ab | 0.08 b | 0.13 a | 0.21 ab | 0.04 bc | 0.04 bc | 0.25 bc |
| 2018 P (%) | | | | | | | | | | |
| $N_0$ | 0.28 a | 0.28 a | 0.19 a | 0.25 a | 0.11 a | 0.19 a | 0.20 a | 0.05 b | 0.12 b | 0.27 ab |
| $CF_{50}$ | 0.28 a | 0.28 a | 0.18 ab | 0.24 ab | 0.10 a | 0.18 ab | 0.19 a | 0.05 b | 0.12 ab | 0.26 b |
| $CF_{100}$ | 0.26 a | 0.26 b | 0.19 a | 0.24 ab | 0.10 ab | 0.18 ab | 0.19 a | 0.05 ab | 0.14 ab | 0.27 ab |
| $CF_{50}PM_{50}$ | 0.26 a | 0.27 ab | 0.19 ab | 0.25 a | 0.10 a | 0.20 a | 0.20 a | 0.06 a | 0.16 a | 0.28 a |
| $CF_{50}CM_{50}$ | 0.27 a | 0.27 ab | 0.18 ab | 0.24 ab | 0.10 a | 0.20 a | 0.20 a | 0.05 b | 0.13 ab | 0.26 b |
| $CF_{50}CP_{50}$ | 0.27 a | 0.27 ab | 0.17 b | 0.24 b | 0.09 b | 0.18 b | 0.19 a | 0.05 b | 0.12 b | 0.28 a |
| 2017 K (%) | | | | | | | | | | |
| $N_0$ | 2.69 c | 2.64 b | 1.81 c | 1.52 c | 0.71 b | 0.74 b | 0.48 b | 0.33 c | 0.87 b | 0.30 ab |
| $CF_{50}$ | 2.77 ab | 2.61 b | 1.95 bc | 1.63 bc | 0.73 b | 0.87 a | 0.47 b | 0.41 bc | 0.84 bc | 0.29 b |
| $CF_{100}$ | 2.73 b | 2.86 ab | 2.04 ab | 1.76 abc | 0.80 ab | 0.75 b | 0.56 a | 0.46 b | 0.82 c | 0.30 ab |
| $CF_{50}PM_{50}$ | 2.96 a | 3.26 a | 2.19 a | 2.03 a | 0.92 a | 0.86 a | 0.51 a | 0.56 a | 0.95 a | 0.36 a |
| $CF_{50}CM_{50}$ | 2.80 ab | 3.05 ab | 2.10 ab | 2.07 a | 0.87 ab | 0.85 a | 0.53 a | 0.43 bc | 0.91 a | 0.31 ab |
| $CF_{50}CP_{50}$ | 2.86 a | 2.73 b | 2.07 ab | 1.89 ab | 0.85 ab | 0.86 a | 0.52 a | 0.40 bc | 0.91 a | 0.29 b |
| 2018 K (%) | | | | | | | | | | |
| $N_0$ | 2.39 a | 2.63 a | 1.71 c | 1.66 ab | 0.92 ab | 0.99 ab | 0.58 a | 0.51 ab | 0.98 b | 0.36 a |
| $CF_{50}$ | 2.35 a | 2.54 ab | 1.78 b | 1.65 ab | 0.91 b | 0.93 b | 0.49 b | 0.48 b | 1.02 ab | 0.32 a |
| $CF_{100}$ | 2.23 a | 2.30 b | 1.78 b | 1.60 b | 0.89 c | 0.93 b | 0.50 ab | 0.47 b | 1.01 ab | 0.31 a |
| $CF_{50}PM_{50}$ | 2.38 a | 2.52 ab | 1.85 a | 1.69 a | 0.93 a | 1.13 a | 0.50 ab | 0.61 a | 1.12 a | 0.32 a |
| $CF_{50}CM_{50}$ | 2.36 a | 2.76 a | 1.86 a | 1.80 a | 0.98 a | 1.11 ab | 0.53 ab | 0.52 ab | 1.09 a | 0.31 a |
| $CF_{50}CP_{50}$ | 2.36 a | 2.75 a | 1.82 a | 1.76 a | 0.96 a | 1.07 ab | 0.52 ab | 0.52 ab | 1.17 a | 0.34 a |

Means followed by the same letter in each year in each column are not significantly different in Tukey's HSD tests ($p < 0.05$).

### 3.3. N, P, and K Accumulation in Rice Plants at Harvest

At harvest time, $CF_{50}PM_{50}$ plants accumulated the most N, at 132.68 and 144.78 kg ha$^{-1}$ in 2017 and 2018, respectively ($p < 0.05$; Table 4). N accumulation was similar between $CF_{50}CM_{50}$ and $CF_{100}$ plants but lower in $CF_{50}CP_{50}$ plants in 2017. Plants treated with organic fertilizers accumulated more N in 2018 than in 2017 (Table 4). N accumulation levels were similar among $CF_{100}$, $CF_{50}CM_{50}$, and $CF_{50}CP_{50}$ plants. Plants treated with $CF_{50}$ accumulated low N amounts in both years.

**Table 4.** N, P and K accumulation (kg ha$^{-1}$) of rice variety Manawthukha affected by organic fertilizer at harvest.

| Treatment | Accumulation (kg ha$^{-1}$) | | |
|---|---|---|---|
| | **N** | **P** | **K** |
| | 2017 | | |
| $N_0$ | 65.6 d | 12.0 d | 50.9 d |
| $CF_{50}$ | 83.4 cd | 15.9 c | 65.0 cd |
| $CF_{100}$ | 125.1 ab | 20.9 ab | 81.9 b |
| $CF_{50} PM_{50}$ | 132.7 a | 21.8 a | 99.4 a |
| $CF_{50} CM_{50}$ | 100.8 bc | 17.5 bc | 74.1 bc |
| $CF_{50} CP_{50}$ | 90.8 cd | 15.8 cd | 66.0 c |
| | 2018 | | |
| $N_0$ | 56.8 b | 12.9 c | 59.6 c |
| $CF_{50}$ | 78.1 b | 16.9 b | 77.1 bc |
| $CF_{100}$ | 123.4 a | 26.1 a | 92.0 ab |
| $CF_{50} PM_{50}$ | 144.8 a | 28.7 a | 110.1 a |
| $CF_{50} CM_{50}$ | 121.5 a | 25.3 a | 96.2 ab |
| $CF_{50} CP_{50}$ | 129.9 a | 25.5 a | 102.3 ab |

Means followed by the same letter in each year in each column are not significantly different in Tukey's HSD tests ($p < 0.05$).

$CF_{50}PM_{50}$ plants also accumulated the most P, at 21.84 and 28.70 kg ha$^{-1}$ in 2017 and 2018, respectively ($p < 0.05$; Table 4). P accumulation was low in $CF_{50}CM_{50}$ and $CF_{50}CP_{50}$ plants in 2017, but similar to that in $CF_{100}$ plants in 2018. Plants treated with organic fertilizers accumulated more P in 2018 than in 2017. Plants treated with $CF_{50}$ accumulated low amounts of P in both years.

Again, $CF_{50}PM_{50}$ plants accumulated the most K, at 99.43 and 110.05 kg ha$^{-1}$ in 2017 and 2018 ($p < 0.05$; Table 4). K accumulation was significantly lower in $CF_{100}$ plants than in plants in other organic fertilizer treatments in both years. $CF_{50}CM_{50}$ and $CF_{50}CP_{50}$ accumulated low amounts of K in 2017, but these amounts increased in 2018. Plants treated with $CF_{50}$ accumulated low amounts of K in both years.

*3.4. HI, Yield, and Yield Characteristics*

In 2017, the highest HI values observed were 0.48, 0.49, and 0.49 for $CF_{50}PM_{50}$, $CF_{50}CM_{50}$, and $CF_{50}CP_{50}$ plants, respectively ($p < 0.05$; Table 5). The highest HI values in 2018 were observed for plants in the same treatments, at 0.51, 0.51, and 0.49, respectively. HI values were low for both years with the sole application of CF.

$CF_{50}PM_{50}$ plants produced the highest number of panicles in 2017 (17.67) and 2018 (20.50; Table 5). Few panicles were produced by $CF_{50}CM_{50}$ and $CF_{50}CP_{50}$ plants in 2017, but the panicle number for both treatments increased in 2018 to values similar to that of the $CF_{100}$ treatment. $CF_{50}PM_{50}$ plants also produced the highest number of spikelets per panicle, at 121.02 and 123.08 in 2017 and 2018, respectively ($p < 0.01$), followed by $CF_{50}CM_{50}$ and $CF_{50}CP_{50}$ plants (Table 5). The lowest number of spikelets per panicle was observed in $CF_{100}$ plants. The numbers of spikelets per panicle for plants treated with PM, CM, and CP were higher in 2018 than in 2017. The highest percentage of filled grains was observed for $CF_{50}PM_{50}$ plants in 2017 and 2018, at 78.55% and 91.40%, respectively ($p < 0.01$; Table 5). Conversely, the lowest percentage of filled grain was observed for $CF_{50}CM_{50}$ and $CF_{50}CP_{50}$ plants in 2017. However, these values were higher in 2018 and similar to the value observed for $CF_{100}$ plants in the same year. No significant differences in thousand grain weights were observed among CF, PM, CM, and CP treatments in 2017 or 2018 (Table 5). In addition, the longest panicle lengths of 25.00 and 24.27 cm were observed in $CF_{50}PM_{50}$ plants in both years. Organic fertilizers generally had a slight effect on panicle length.

Rice treated with $CF_{50}PM_{50}$ produced the highest yields, at 6.90 and 7.42 t ha$^{-1}$ in 2017 and 2018, respectively ($p < 0.01$; Table 5). $CF_{100}$ plants produced similarly low yields of 6.30 and 6.70 t ha$^{-1}$

in 2017 and 2018, respectively. The yields were significantly higher in all crops treated with organic fertilizers than in those treated with $CF_{100}$ in 2018. $CF_{50}CM_{50}$ and $CF_{50}CP_{50}$ plants produced yields of 7.32 and 7.11 t ha$^{-1}$. $CF_{50}$ plants did not produce a high yield.

**Table 5.** Harvest index, yield and yield attributes of rice variety Manawthukha affected by organic fertilizer.

| Treatment | Harvest Index | No. Panicle Hhill$^{-1}$ | No. Spikelets Panicle$^{-1}$ | Filled Grain (%) | 1000 Grain Weight (g) | Max. Panicle Length (cm) | Yield (t ha$^{-1}$) |
|---|---|---|---|---|---|---|---|
| | | | 2017 | | | | |
| $N_0$ | 0.37 b | 13.7 b | 66.9 b | 67.5 d | 19.5 a | 15.7 b | 3.8 c |
| $CF_{50}$ | 0.33 c | 14.7 ab | 65.6 b | 73.9 bc | 20.6 a | 24.2 a | 4.5 b |
| $CF_{100}$ | 0.45 ab | 17.3 a | 102.8 a | 77.2 ab | 19.2 ab | 24.5 a | 6.3 ab |
| $CF_{50}PM_{50}$ | 0.48 a | 17.7 a | 121.0 a | 78.6 a | 21.0 a | 25.0 a | 6.9 a |
| $CF_{50}CM_{50}$ | 0.49 a | 14.8 ab | 118.3 a | 71.4 cd | 18.2 ab | 24.3 a | 5.9 ab |
| $CF_{50}CP_{50}$ | 0.49 a | 14.2 b | 106.7 a | 72.5 c | 18.4 ab | 24.7 a | 5.5 ab |
| | | | 2018 | | | | |
| $N_0$ | 0.36 c | 13.3 c | 67.2 b | 74.2 b | 17.3 a | 21.5 b | 3.1 c |
| $CF_{50}$ | 0.38 bc | 16.8 b | 58.6 b | 86.1 a | 18.7 a | 23.2 ab | 4.3 b |
| $CF_{100}$ | 0.46 ab | 17.7 ab | 103.8 a | 88.7 a | 18.5 a | 23.8 ab | 6.7 ab |
| $CF_{50}PM_{50}$ | 0.51 a | 20.5 a | 123.1 a | 91.4 a | 18.9 a | 24.3 a | 7.4 a |
| $CF_{50}CM_{50}$ | 0.51 a | 17.2 b | 119.3 a | 87.6 a | 18.5 a | 23.4 ab | 7.4 a |
| $CF_{50}CP_{50}$ | 0.49 a | 17.2 b | 107.2 a | 89.0 a | 18.5 a | 23.5 ab | 7.1 a |

Means followed by the same letter in each year in each column are not significantly different in Tukey's HSD tests ($p < 0.05$).

## 4. Discussion

Rice plants have to absorb enough N, P, and K during the growth stage to obtain optimum growth characteristics and yield. CFs have been widely used to enhance rice yield. However, the application of large amounts of CF has led to the degradation of agricultural land and declining crop yields. Therefore, we applied well-decomposed organic fertilizers (PM, CM, and CP) based on their EMN to experimental plots to supply sufficient N, P, and K to the rice grown in these plots. Organic fertilizers have various nutrient composition, especially total N content. The PM contains a higher total N ($\geq$4%) than the CM (<4%) and CP (<4%). The PM applied lower total N and total weight than those of CM and CP to adjust the same EMN level (42.5 kg ha$^{-1}$) because the EMN for PM (total N $\geq$4%) is 50% and for both CM (total N <4%) and CP (total N <4%) is 30% in the first year [13]. And, the PM has a lower C:N ratio (6.8) than CM (15.23) and CP (17.71). The responses to rice plants were different among organic fertilizers using EMN method.

Initially, plants treated with $CF_{100}$ produced more DM than plants treated with organic fertilizers. The CF used is readily soluble and hence can supply nutrients to rice plants within a short time after application [26]. However, SPAD values, tiller numbers, and plant heights were not statistically different between $CF_{100}$ plants and $CF_{50}PM_{50}$, $CF_{50}CM_{50}$, and $CF_{50}CP_{50}$ plants. The leaf and sheath N, P, and K contents of plants grown with $CF_{100}$ were higher than those of $CF_{50}CM_{50}$ and $CF_{50}CP_{50}$ plants but lower than those of $CF_{50}PM_{50}$ plants in the tillering stage. After this stage, growth characteristics did not differ between $CF_{100}$ and $CF_{50}PM_{50}$ plants. Organic fertilizers release nutrients slowly [9]; thus, rice plants treated with these fertilizers may be nutrient-deficient early in the growth stage. However, we did not observe any symptoms of nutrient deficiency in the rice plants in this study. The optimum N, P, and K contents in plant tissues from the tillering to panicle-initiation stages are 2.9%–4.2%, 0.2%–0.4%, and 2.3%–2.9%, respectively [27]. Leaf and sheath N, P, and K contents were higher in $CF_{50}PM_{50}$ plants than in the active tillering and panicle-initiation stages in both years.

The development of the tiller primordium depends on the N, P, and K contents in leaves and sheaths. Tiller number increases linearly with sheath N content [25]. A high sheath N content increases the cytokinin content within tiller nodes and enhances the germination of the tiller primordium [28]. Protein and chlorophyll synthesis are also related to leaf N content; higher photosynthetic rates result in greater stem elongation and leaf area expansion [29]. The highest SPAD values, tallest plants, and



greatest tiller numbers were recorded for the $CF_{50}PM_{50}$ plants. PM contains more readily available micro- and macro-nutrients, which are more easily absorbed by plants [30]. Thus, the application of PM may have enhanced plant growth characteristics in our study.

By contrast, low SPAD values, few tillers, and short plants were observed in the $CF_{50}CM_{50}$ and $CF_{50}CP_{50}$ treatments before the panicle-initiation stage. $CF_{50}CM_{50}$ and $CF_{50}CP_{50}$ plants also produced the lowest amounts of DM, similar to $CF_{100}$ plants. These results could be attributable to the low leaf and sheath N, P, and K contents. The amount of nutrients released from organic fertilizers, which determines nutrient availability and uptake, depends on the C:N ratio or total N content [31]. In our study, CM and CP contained less total N and higher C:N ratio than PM.

For rice, the N, P, and K amounts absorbed during the panicle-initiation stage determine panicle primordia formation, panicle branching, and the setting of spikelets [25]. $CF_{50}PM_{50}$ plants had absorbed large amounts of N, P, and K by the end of this stage. Consequently, $CF_{50}PM_{50}$ plants had the highest panicle number and longest panicles. Panicle length increased in tandem with sheath N content [32]. In addition, the flowering and grain-filling stages are critical for rice yields. If plants are nutrient-deficient, particularly in N, spikelet number would decrease [25]. The $CF_{50}PM_{50}$ treatment maintained high N, P, and K contents in leaves and sheaths. Then these nutrients were translocated to help develop the panicles. Thus, the percentage of filled grains was highest in the $CF_{50}PM_{50}$ treatment.

$CF_{50}PM_{50}$ plants had the highest SPAD value for flag leaves and the highest N, P, and K contents in leaves and sheaths, except during the harvest stage. Seed N, P, and K contents were also highest in $CF_{50}PM_{50}$ plants. Rice yield was positively correlated with seed N and P contents [27]. K content did not affect tillering, the number of spikelets per panicle, percentage of filled grains, or grain weight. Sheath, leaf, and seed K contents help to improve the tolerance of rice plants to adverse climatic conditions, lodging, insect pests, and diseases [33]. The plants only responded to K in terms of yield when N and P were sufficiently available. $CF_{50}PM_{50}$ plants accumulated the most total N, P, and K and produced the highest yields in 2017 and 2018. The PM supplied enough nutrients during the vegetative stage to ensure a good yield. If rice plants can absorb sufficient N, P, and K before the panicle-initiation stage, they would produce a good yield even if nutrients were in short supply later [25]. For example, in previous studies, the application of both organic manure and CF instead of CF only led to significantly higher grain yields, as enough N, P, and K were supplied in the former treatment [34,35].

Organic fertilizers provide a more balanced mix of nutrients to plants, particularly micronutrients, which improve rice yields [36]. Seed N and P contents were high in all treatments using organic fertilizers in 2018. Total N, P, and K contents and yields in treatments using PM, CM, and CP were also significantly higher in 2018 than in 2017. Compared to yields of $CF_{100}$ plants, the yields of $CF_{50}PM_{50}$ plants were 8.69% and 9.70% higher in 2017 and 2018, respectively. Due to the higher grain yields, higher HI values were recorded for all treatments using organic fertilizers. Higher yields and HI values indicate more efficient partitioning of photosynthetic products to economic yields [37]. By contrast, yields of $CF_{100}$ plants remained similar over both years, with correspondingly low HI values.

Each organic fertilizer had a different effect on yield in both years when applied based on EMN. We calculated the yield that was solely dependent on organic fertilizers for the Manawthukha rice variety (Figure 5). In 2017, 43.68% of the total yield was attributed to PM (total N > 4%), 33.85% to CM (total N < 4%), and 29.75% to CP (total N < 4%). Similarly, 42.33%, 41.97%, and 39.81% of the total yield was attributed to PM, CM, and CP, respectively, in 2018. PM, which contains total N (> 4%), supplied a more balanced mix of NPK nutrients that was synchronized to the needs of the rice plants throughout the crop period in both years. The application of CM (total N <4%) and CP (total N <4%) only led to higher yields than in the $CF_{100}$ treatment in 2018. Manure with total N ≥ 4% releases 50% of its mineralizable N in the first year after application [13]. Conversely, organic matter with a low nitrogen content decomposes slowly and acts like a continuously applied fertilizer. For example, in a previous study, CM and CP affected rice growth and yield and soil fertility for 3 years after initial application [38]. Therefore, the characteristic of organic fertilizer, especially total N is important for the purpose of the application.

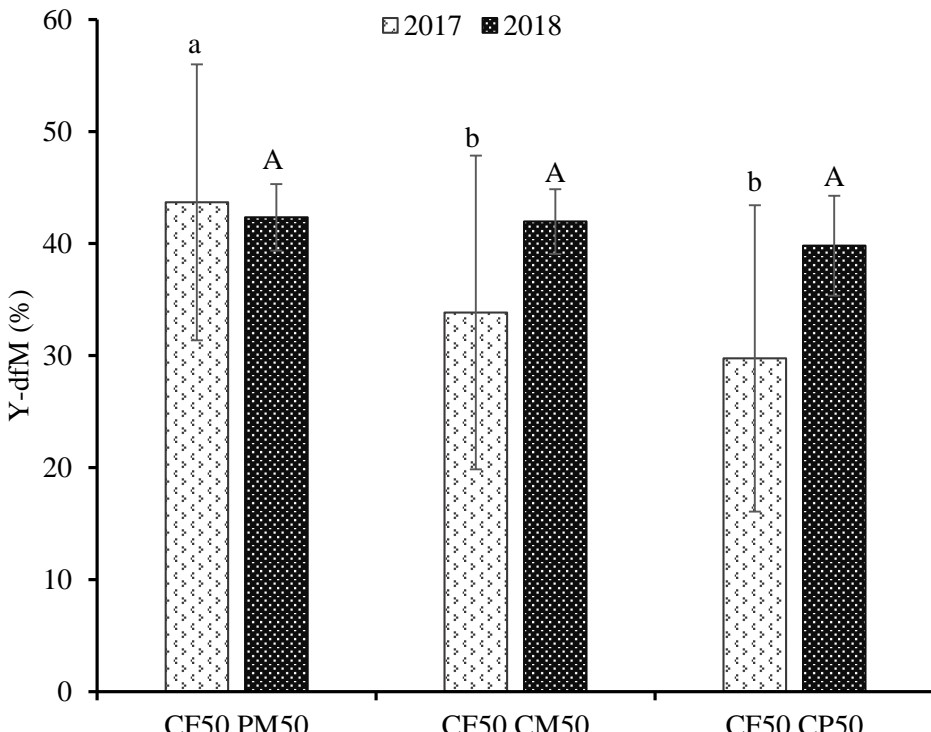

**Figure 5.** Yield of rice variety Manawthukha derived from manure (Y-dfM; %) as affected by organic fertilizers as EMN method. The histograms with the same letter in the same case in each year are not significantly different by the Tukey HSD test ($p < 0.05$). Y-dfM (%) = Yield [(50% organic + 50% inorganic) – 50% inorganic] ÷ Yield (50% organic + 50% inorganic) × 100. Vertical bars represent SD of 3 samples.

Farmers in developing countries need fertilization techniques that are easier, cheaper, and more efficient than using CFs. In this study, we proposed that organic fertilizers can be applied according to their EMN contents, which were calculated based on total N contents. The EMN equation is also easy for farmers to calculate (see Materials and Methods section). This method allows the nutrients supplied from organic fertilizers to be synchronized to the nutrient demand of the rice crop, which results in higher yields. The increased use of organic fertilizers can lead to reduced CF usage and thus less environmental pollution. Furthermore, synchronizing N supply to demand ensures that only the optimum amount of organic fertilizers is used. By contrast, applying organic fertilizers based on their total weight does not account for the N demand by the rice crop.

## 5. Conclusions

To the best of our knowledge, this is the first study to comprehensively evaluate how applying organic fertilizers using the EMN method affects NPK status, growth characteristics, and yield of the Indica-type, Manawthukha rice variety. Organic fertilizers with total N ≥ 4% had higher availability of N that was synchronized to the N demand of the rice crop. P and K availability was also high in both experimental years. The co-application of 50% EMN from organic fertilizer with total N ≥ 4% and 50% N from CF led to higher yields than those observed in the $CF_{100}$ treatment in the first as well as second years. Organic fertilizers with total N < 4% were only effective at improving growth and yield in the second year, after being continuously applied for 2 years. Overall, we recommend the EMN method for the application of organic fertilizer in rice cultivation. This method not only effectively fertilizes the rice crop but is also a sustainable way of producing rice. Substituting CFs with organic alternatives would also help protect the environment.

**Author Contributions:** K.M. performed field experiments for over 2 years, analysis data, interpret the results and wrote the whole manuscript. A.Z.H. and T.T.P.T. participated in the data collection in every week. Y.K. helped for the land preparation, and irrigation. T.Y. supervised this research, suggested the data analysis, reviewed the manuscript and gave valuable comments.

**Funding:** This study was supported by the Japanese Government (MEXT) Scholarship Program 2016–2019, Japan.

**Acknowledgments:** We acknowledge the technical staff of the University Farm, Kyushu University for supporting the fieldworks in field experiments.

**Conflicts of Interest:** We have disclosed that there is no conflict of interest regarding the publication of this article.

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
