# Peer review of "Effects on NPK Status, Growth, Dry Matter and Yield of Rice (Oryza sativa) by Organic Fertilizers Applied in Field Condition"

_agriculture, doi:10.3390/agriculture9050109_

Round 1
Reviewer 1 Report
The manuscript (MS) evaluated investigated the N, P and K contents of plants parts (leaf, sheath, panicle and seed) of rice affected by organic fertilizer throughout the crop period. The theme of this manuscript is within the scope of Agriculture and including valuable data. However, there are some fundamental concerns on this MS (especially for introduction and discussion, and for details, please see the following comments). So, I think some revisions which can address the concern are needed for considering acceptance of this MS. Please consider the comments below for your revision of the MS.[Introduction]
Importance of this study is not clear. Although the introduction listed so much literatures, rationale or question driving the study is not clear to me. That is, although a lot of general knowledge is stated, there are few descriptions linked to this research. A better introduction articulates the question clearly and provides testable hypotheses. Please write a reason why you need to carry out this study. “Few research” is not a reason. There is a reason, but no (or few) research, therefore we have to conduct it. In addition, the logical process leading to the hypothesis described in L86-88 is not clear. Please provide a suitable hypothesis with reasonable explanation.
[M & M]
L94: “clay loam” is “soil texture”, not “soil type”.
Table 1: Is the “weight” of organic fertilizers dry weight or fresh weight?
Table 2: C:N ration of organic fertilizers should be added to discuss their decomposability.
L144: “Dr. Umezaki, Mie University” -> “Myint et al. (2011)”
L159: Please describe what the SPAD value is an indicator for.
Figure 1-3: SD of three plots (average of five hills per plot) should be presented.
Figure 2 and 3 are shown in reverse each other. Please check it.
Figure 3 (No. of tillers): There is no legend for CF50 CP50 in the figure.
[Results]
L263: What is the standard value of the “optimum N”?
Table 4 and 5: There are too much significant figures.
Table 5: Please correct the items for N0 shown in bold to fine, and remove their underline.
[Discussion]
There is a significant lack of discussion about the effect of continuous application of organic fertilizers (only L373-377). The effects of organic fertilizers application on rice growth (and soil fertilities) continue for a long time (not a single year) (e.g., Nishida et al. 2008, Soil Sci. Plant Nutr., 54, 459–466). Therefore, when determining the application rate of organic fertilizers assuming continuous use, it is necessary to carefully consider the continuous effect. The effects of continuous application should be described also in the conclusions and the abstract.
L385: I disagree. Application of organic matter also supplies NPK to farmland (sometimes excessive application), resulting in environmental pollution. 
[Conclusions]
Is the thorough investigation on NPK needed to draw the conclusions? Is it enough to investigate only N?
[References]
“In Japanese” or “In Japanese with English summary” should be added to the articles in Japanese.
Sincerely yours,
Author Response
Reviewer 1
[Introduction]
Point 1: Importance of this study is not clear. Although the introduction listed so much literatures, rationale or question driving the study is not clear to me. That is, although a lot of general knowledge is stated, there are few descriptions linked to this research. A better introduction articulates the question clearly and provides testable hypotheses. Please write a reason why you need to carry out this study. “Few research” is not a reason. There is a reason, but no (or few) research, therefore we have to conduct it. In addition, the logical process leading to the hypothesis described in L86-88 is not clear. Please provide a suitable hypothesis with reasonable explanation.
Response 1; We revised the introduction. We gave the reason for carrying out this study in L 61 - 74. And, we edited the hypothesis with explanations in L 87 - 91. Then, we checked English grammar of the manuscript with Textchecker.com. The certificate is attached.
[M & M]
Point 2: L94: “clay loam” is “soil texture”, not “soil type”.
Response 2; We correct it in L 98-99.
Point 3: Table 1: Is the “weight” of organic fertilizers dry weight or fresh weight?
Response 3; We add dry weight in Table 1.
Point 4: Table 2: C:N ration of organic fertilizers should be added to discuss their decomposability.
Response 4; We added it Table 2.
Point 5: L144: “Dr. Umezaki, Mie University” -> “Myint et al. (2011)”
Response 5; We correct it in L 150-151.
Point 6: L159: Please describe what the SPAD value is an indicator for.
Response 6; We add the indicator information about SPAD value in L 165-167.
Point 7: Figure 1-3: SD of three plots (average of five hills per plot) should be presented.
Response 7; We corrected it in figure 1-3.
Point 8: Figure 2 and 3 are shown in reverse each other. Please check it.
Response 8; We corrected it.
Point 9: Figure 3 (No. of tillers): There is no legend for CF50 CP50 in the figure.
Response 9; We corrected it in figure 3.
[Results]
Point 10: L263: What is the standard value of the “optimum N”?
Response 10 The standard value for optimum N uptake of Manawthukha is 110-120 kg ha-1 according to report (unpublished) of Department of Agricultural Research, Myanmar. But, we correct the sentence in L 285.
Point 11: Table 4 and 5: There are too much significant figures.
Response 11; We want to thoroughly highlight the effects of organic fertilizer as EMN in terms of major nutrients accumulation as well as yield and yield attributes as a complete story.
Point 12: Table 5: Please correct the items for N0 shown in bold to fine, and remove their underline.
Response 12; We corrected it in table 5.
[Discussion]
Point 13: There is a significant lack of discussion about the effect of continuous application of organic fertilizers (only L373-377). The effects of organic fertilizers application on rice growth (and soil fertilities) continue for a long time (not a single year) (e.g., Nishida et al. 2008, Soil Sci. Plant Nutr., 54, 459–466). Therefore, when determining the application rate of organic fertilizers assuming continuous use, it is necessary to carefully consider the continuous effect. The effects of continuous application should be described also in the conclusions and the abstract.
Response 13; We discussed the effects of continuous application of organic fertilizer in L 400-412. We add it in the conclusion and abstract, too.
Point 14: L385: I disagree. Application of organic matter also supplies NPK to farmland (sometimes excessive application), resulting in environmental pollution. 
Response 14; We correct it in L 447-448.
[Conclusions]
Point 15: Is the thorough investigation on NPK needed to draw the conclusions? Is it enough to investigate only N?
Response 15; We edit the conclusion in L 453-454.
[References]
Point 16: “In Japanese” or “In Japanese with English summary” should be added to the articles in Japanese.
Response 16; We correct it in references.
Reviewer 2 Report
Dear Colleagues,
the manuscript report some informations about the organic fertilization of rice. This kind of research is not really innovative, but some data are probably in interest of readers of "Agronomy".
The limits of the present study are: (i) the number of years of experimentation, only two years; (ii) are the same plots used for the trial both years? It is not clear; (iii) the characterization of organic fertilizers is scarce, organic fertilizers are not inorganic (and well defined) fertilizers, a lot of informations are needs in order to understand your effect of soil and plants nutrition.
A lot of minor remarks are reported in the file included.
Sincerely yours.
Author Response
Reviewer 2
Point 1: The limits of the present study are: (i) the number of years of experimentation, only two years;
Response 1: (i) The number of years of experimentation is two years during my Ph D study but the significant effects of organic fertilizer as EMN on growth and yield of rice were observed.
Point 2: (ii) are the same plots used for the trial both years? It is not clear;
Response 2: (ii) The same plots are used for the trials in both years. The position of each plot of treatments did not change to observe the continuous effects of organic fertilizer in both years, mentioned in L 118-119.
Point 3: (iii) the characterization of organic fertilizers is scarce, organic fertilizers are not inorganic (and well defined) fertilizers, a lot of informations are needs in order to understand your effect of soil and plants nutrition.
Response 3: (iii) the characterization of organic fertilizers is mentioned in Table 2 and L 337-342. We highlighted the differences between organic and inorganic fertilizer in the introduction (L 62-67). Among the characteristics of organic fertilizer, we mainly focused the total N content that affects significantly the mineralizable N supply from organic fertilizer, which in turn influence the growth and yield of rice. Therefore, organic fertilizer application as EMN had the significant effect on nutrient accumulation, growth and yield of rice.
Point 4: A lot of minor remarks are reported in the file included.
Response 4: Minor remarks are deleted.
Reviewer 3 Report
I suggest to search more recent Reference notes for Introduction section;
correct fig 1,2, and 3.

Author Response
Reviewer 3
Point 1: I suggest to search more recent Reference notes for Introduction section;
Response 1: We added the recent references as suggested and highlighted in yellow color for the changes.
Point 2: Correct fig 1,2, and 3.
Response 2: We correct figure 1, 2 and 3.
Round 2
Reviewer 1 Report
 Thank you for your reply. I agreed most of your explanation and revisions. Please consider the comments below for your revision of the manuscript.
L68: Because the N release from organic fertilizers continues over several years, the duration of release rate (20%, 30% and 50%) should be noted. “One crop season” or “a single year”?
Table 4 and 5: As I pointed out on the first version, there are too much significant figure (significant digit). I meant that there is too much significant figure for each item. For most items, three digits would be appropriate (e.g.: Number of tillers: 13.67 -> 13.7). Significant figures indicate the accuracy of the measurement. Improper description reduces the reliability of the results. So, please use appropriate significant figure for each item.
L337-342: I could not understand what the author means. Please rewrite it more clearly.
Figure 5: The results of statistical analysis of 2018 (A, A, AB) are wrong. If there is no significant difference among the treatment, all treatments should be “A”. Besides, please describe about error bar in the legend.
Sincerely yours,
Author Response
Point 1: L68: Because the N release from organic fertilizers continues over several years, the duration of release rate (20%, 30% and 50%) should be noted. “One crop season” or “a single year”?
Response 1: We noted it in L 69.
Point 2: Table 4 and 5: As I pointed out on the first version, there are too much significant figure (significant digit). I meant that there is too much significant figure for each item. For most items, three digits would be appropriate (e.g.: Number of tillers: 13.67 -> 13.7). Significant figures indicate the accuracy of the measurement. Improper description reduces the reliability of the results. So, please use appropriate significant figure for each item.
Response 2: We corrected three digits in Table 4 and 5. Thanks for your very good suggestion. We learnt a new knowledge.
Point 3: L337-342: I could not understand what the author means. Please rewrite it more clearly.
Response 3: We rewrote it in L 337 and 343.
Point 4: Figure 5: The results of statistical analysis of 2018 (A, A, AB) are wrong. If there is no significant difference among the treatment, all treatments should be “A”. Besides, please describe about error bar in the legend.
Response 4: We corrected (A, A, A), the typing mistake in the results of 2018 in Figure 5. We described about error bar in the legend.